# Emerging Risk of Cross-Species Transmission of Honey Bee Viruses in the Presence of Invasive Vespid Species

**DOI:** 10.3390/insects14010006

**Published:** 2022-12-21

**Authors:** María Shantal Rodríguez-Flores, Maurizio Mazzei, Antonio Felicioli, Ana Diéguez-Antón, María Carmen Seijo

**Affiliations:** 1Department of Plant Biology and Soil Sciences, University of Vigo, Campus As Lagoas, 32004 Ourense, Spain; 2Department of Veterinary Sciences, University of Pisa, Viale delle Piagge 2, 56124 Pisa, Italy

**Keywords:** viruses, bee, vespidae, *Vespa velutina*

## Abstract

**Simple Summary:**

Currently, bee viruses are one of the threats to honey bee populations. The increase in invasive Vespid species causes a negative impact on honey bee colonies, weakening them and making them more vulnerable to the presence of pathogens. This bibliographic review shows the main research on bee viruses in the *Vespa*, *Vespula* and *Polistes* genera belonging to the Vespidae family. This taxon contains several of the most widespread invasive Vespid species worldwide. Many of these species are predators of honey bees and cause great impacts, as is the case with the yellow-legged hornet *Vespa velutina*. The presence of viruses in species of Vespids that interact with honey bees could represent an emerging risk in the transmission of pathogens, weakening the defense strategies of native species. Gathering this information is necessary to promote the research on the spread of bee viruses associated with invasive species as well as in the development of bee virus control strategies.

**Abstract:**

The increase in invasive alien species is a concern for the environment. The establishment of some of these species may be changing the balance between pathogenicity and host factors, which could alter the defense strategies of native host species. Vespid species are among the most successful invasive animals, such as the genera *Vespa*, *Vespula* and *Polistes*. Bee viruses have been extensively studied as an important cause of honey bee population losses. However, knowledge about the transmission of honey bee viruses in Vespids is a relevant and under-researched aspect. The role of some mites such as *Varroa* in the transmission of honey bee viruses is clearer than in the case of Vespidae. This type of transmission by vectors has not yet been clarified in Vespidae, with interspecific relationships being the main hypotheses accepted for the transmission of bee viruses. A majority of studies describe the presence of viruses or their replicability, but aspects such as the symptomatology in Vespids or the ability to infect other hosts from Vespids are scarcely discussed. Highlighting the case of *Vespa velutina* as an invader, which is causing huge losses in European beekeeping, is of special interest. The pressure caused by *V. velutina* leads to weakened hives that become susceptible to pathogens. Gathering this information is necessary to promote further research on the spread of bee viruses in ecosystems invaded by invasive species of Vespids, as well as to prevent the decline of bee populations due to bee viruses.

## 1. Vespid Species as an Invasive Alien Species

Globalization is causing decisive changes in human habitats and ecosystems. The introduction of invasive species and the appearance of infectious diseases in certain populations are well known processes in this field. It was estimated that the economic cost of invasive species has been $1.288 trillion over the past 50 years [1], including many pathogens that have direct impacts on human activities and the health of species worldwide.

The success of invasive species depends on several factors, but the most recent findings suggest that the success of the invasion may depend on their ability to respond to natural selection, highlighting the importance of their genetic architecture, leading to adaptations to new environments [2]. Furthermore, the absence of enemies in an area where the exotic species are introduced may contribute to pest species settling in. This is known as “The enemy release hypothesis”. This could be explained by a decrease in enemy diversity in the invasive area compared to the native area, or by the fact that non-native species are less affected by enemies than native species in the invaded community [3]. Moreover, when species move to new environments, the probability of transporting the pathogens that interact with them is low, since they generally experience a bottleneck that affects survival [4]. Despite this, the increase in the expansion of exotic species could be influencing the global spread of pathogens, since these species and infectious diseases follow similar patterns, especially the greater connection and globalization [5].

The Vespidae family is vastly widespread and comprises almost all species of eusocial wasps, as well as many solitary species [6,7]. Although the proportion of species that successfully settle in new areas is low, the solitary and social wasps are a very favorable group to be an invasive species and, in many cases, they are considered pests. Many Vespids alter the composition of ecosystems since, being generalist predators, they change interactions or cause complex interactions with the host community [8]. The social wasps from the genera *Vespula, Vespa*, and *Dolichovespula* are pests in temperate and tropical regions of the world [9]. Two of the most abundant species are *Vespula vulgaris* (*V. vulgaris)* and *Vespula germanica (V. germanica)*. These two species show a very plastic generalist foraging behavior, which favors the appearance of a great variety of impacts on biodiversity and changes in the function of the ecosystems invaded [7,10,11]. The case of *Vespa* is relevant since some species such as *Vespa velutina (V. velutina)* or *Vespa orientalis (V. orientalis)* are species of concern for ecosystems and human health, when introduced as invasive species [7]. *V. velutina nigrithorax* is the only one included in the List of Invasive Alien Species of Union concern (Regulation (EU) 1143/2014 of the European Parliament and of the Council of 22 October 2014 on the prevention and management of the introduction and spread of invasive alien species) and today it is a widespread pest in Europe. 

*V. velutina* or Asian yellow-legged hornet is common in central to eastern Asia [12,13,14,15]. In their natural range, populations of *V. velutina* are kept in balance with the species they prey on (Asian honey bee *Apis cerana* (*A. cerana*)) and with the pathogens that are involved in their biological cycle [8]. The species was detected for the first time in France in 2004, and since then it has been slowly spreading throughout the rest of Europe [7,14,16]. One of the main impacts is due to its high capacity for predation of multiple species of insects, especially of the honey bee, altering the beekeeping sector at all levels [14,16]. In addition, this invasive insect is very flexible in its diet, since in addition to preying on an enormous variety of insect species, it also consumes other small animals, carrion from animal carcasses, and other resources such as sugary substances from nectar or fruits. This wide and diverse diet makes it a risk factor since it gives rise to a wide spectrum of interactions in the invaded ecosystems.

Knowledge about pathogens could be crucial to better understand the ecology of pest species and thus develop an integrated control strategy. Henceforth, this study aims to deepen the knowledge about the most common bee viruses found in different Vespid species. Additionally, this review makes a special mention of the *V. velutina* hornet, as it is a species that is expanding rapidly in all European ecosystems, thus causing a notable environmental and socioeconomic impact, especially in the beekeeping sector [14,16]. The role of this invasive species in the spillover of potential pathogens is still not well known. However, since a higher number of interrelations between species increases the risk, understanding the factors that facilitate the transmission is essential to establish control strategies. 

## 2. Vespid Species as Hosts of Bee Viruses

Most invasive Vespid species interact extensively with other species, when preying on other insects such as honey bees, and during foraging and the search for resources to build nests. The interaction between these species allows the transmission of many pathogens, including bee viruses [17]. Thus, virus transfer between pest species and target species such as honey bees and vice versa is an expected scenario. Genomic approaches have greatly expanded understanding of the diversity of bee viruses, their routes of transmission, and the strategies that bees have developed to combat viral infections [18]. These infections are further exacerbated by major stressors such as parasites, poor nutrition, or the arrival of new predatory pest species, such as Vespids. The recent confirmation of replicability of bee viruses in some species of Vespids [19,20] could suggest that the interactions between pathogen and newly introduced host are being reconfigured.

In recent years, with the entry of the invasive species *V. velutina* in Europe, more research on viruses such as pathogens has been conducted, searching for the best control method to reduce hornet populations [20,21,22,23]. In parallel, the interest of researchers and beekeepers in the study of honey bee viruses is increasing, mainly due to their association with certain parasites such as *Varroa* [24]. Moreover, the enhancement of diagnostic techniques such as Next Generation Sequencing [19,20,21,25,26] makes it easier to detect these viruses in large sample size. However, so far there is still little knowledge about the presence of bee viruses and their transmission to other hosts such as the different species of hornets and their interrelationships.

In this sense, invasive polyphagous species are more successful in the establishment than monophagous or oligophagous species [20]. The transmission of pathogens is another consequence of the introduction of these invasive species [20]. Although the expansion of some species of invasive Vespids, such as *Vespa* or *Vespula*, to new areas could imply a reduction of its native pathogens, its rapid adaptability to new ecological niches gives rise to new interactions with existing pathogens in the new invaded areas. Other successful invasive insects are several *Polistes* species (*Polistes dominula (P. dominula)* or *Polistes chinensis* (*P. chinensis*)) that can also be considered as a pest in several countries [7,27]. Although species of the genus *Polistes* do not prey on bees, the spillover effect of the different bee viruses in ecosystems makes it a carrier and host for these viruses.

These species affect invaded local biodiversity either through direct predation or competition for food, nest settlements or space. Many invasive social wasp species are also increasingly recognized for their predation on honey bees [4,14,15,16]. It is noteworthy that most of the research dedicated to these invasions appears in the simultaneous phase of the invasion. From the point of view of the literature, much of this research focuses on studying the spread of pathogens between Vespid and honey bee species (hereinafter cited). Most of these apply genomic technologies, and their objectives are mainly focused on managing control strategies for these invasive species [28].

## 3. Types of Viral Transmission

Knowing how viruses spread and cause infections is essential. There are various ways of transmitting a virus. These are summarized in horizontal and vertical transmission, or both. In the horizontal transmission, the virus is usually transmitted between members of the same species that do not have a progenitors–descendants relationship, that is, between individuals of the same generation [29]. In turn, this type of transmission can be direct or indirect. The direct route involves an infection transmitted by environmental contamination, food, feces, or sexual infection. The indirect route involves a transmission vector, an intermediate biological host (such as mosquitoes or mites [24]) that acquires and transmits the virus from one host to another [29]. In general, infection in vertical transmission takes place through the host’s offspring. The virus is transmitted vertically from the parental generation to the brood through the egg. This can occur in two ways, by transmission on the surface of the egg (*transovum* transmission) or inside the egg (*transovarian* transmission) [29,30]. 

Virulence tends to increase in horizontal transmission, unlike in vertical transmission [29]. The higher the level of pathogen multiplication, the greater its virulence; thus, transmission and virulence will be positively correlated [29]. In vertical transmission, the reproduction of the host contributes to the reproduction of the parasite, so this type of transmission exerts a natural selection on the parasites to be less harmful to its hosts since the survival of the one implies the survival of the other [31]. This type of natural selection can eliminate the pathogen unless they develop special routes of transmission such as biparental transmission, in which the virus is transmitted by males (sperm) and females (ovules) to their offspring. Another method of increasing representation in future generations in vertical transmission, and thus balancing the effects of natural selection, is the sex-ratio distortion. This is the case of *Pteromalus puparum* negative-strand RNA virus 1 (PpNSRV-1), which is an RNA virus from a parasitoid wasp (*Pteromalus puparum*). This virus is transmitted vertically by infected wasps, biparentally (both female and male), but at the same time the virus reduces the number of female offspring by modifying the secondary sex ratio of the parasitoid host [32]. Although each type of transmission has certain characteristics to succeed in its reproduction and expansion, there are many viruses spreading through a combination of vertical and horizontal transmission [31].

Deformed Wing Virus (DWV), one of the most widely studied bee viruses, may be an example of both the vertical and horizontal transmission in the honey bee. Numerous studies describe the presence of DWV in honey bee queen ovaries and therefore a venereal–vertical infection [30,33,34,35]. Although it can also be transmitted horizontally [17], vertical transmission has been demonstrated to be the most generalized natural route of spread [30]. This class of viruses can be considered hereditary genetic agents that can cause potential impacts on the ecology and evolution of insects [36]. Apart from that, studies on DWV in Vespids suggest horizontal transmission mainly due to predation on honey bees and other insects [21]. In addition, the presence of DWV has been demonstrated in nectar and pollen of plants [17] that insects forage, which might be an added kind of horizontal transmission. However, this relationship between Vespids and DWV in nectar has not yet been proven. 

The type of viral transmission depends largely on the type of relationship that exists between the interacting species. The relationships between pathogen and wasps can vary from pathogens to mutualists. An example is a relationship between *Ascovirus* DpAV4 and wasps of the genus *Diadromus* [37]. This relationship can be pathogenic, mutualistic or non-pathogenic, depending on the regulatory factors that control virus replication and on the type of relationship that has been established. Many viruses can be beneficial for their hosts, through a symbiotic interaction [38,39]. This is the case of parasitic wasp species of the genus *Diadromus* that harbors reovirus (family Reoviridae). It has been demonstrated that some of these reoviruses can suppress host defense, facilitating the development of parasitoid eggs [40]. In addition, some viral pathogens can influence other parameters such as the size of the nests. Thus, the Kashmir bee virus (KBV) affects the size of the nest in the common invasive wasp *V. vulgaris* [41]. The longevity of wasp colonies has also been influenced by the presence of diverse viral loads. This is the case for colonies of the *Vespula pensylvanica* (*V. pensylvanica*) wasp, whose longevity may be affected by the viral load of the Moku virus [42].

## 4. Transmission Vectors and Activators

### 4.1. Mites

Many of the bee viruses use transmission vectors such as mites. According to De Jong et al. [43], mites that affect honey bees or that are found in hives can be parasites, phoretic mites, and house hosts. Some of the parasitic mite species present in hives can cause serious diseases in honey bees. The main mites that cause problems are the *Varroa* genus, mainly the *Varroa jacobsoni* (*V. jacobsoni*) and *Varroa destructor* (*V. destructor*), *Acarapis woodi*, and *Tropilaelaps mercedesae*. Among them, the *Varroa* is the causative agent of varroosis one of the most serious diseases in honey bees [24]. In addition, it has the potential to contribute to the worldwide mortality of honey bee colonies through several RNA viral infections [44]. One of the most extended viruses associated with this mite is the DWV. Although this virus can appear without the presence of *Varroa*, it has been proven that its presence increases the frequency and quantity of this infection in honey bees [44]. This virus has been detected symptomatically and asymptomatically in some species of Vespidae and some predators of *Apis mellifera* (*A mellifera*), which suggests the possibility that the virus can be transmitted by ingesting infected bees [21,25,41,45,46]. Another virus associated with varroa (*V. destructor*) is the Acute bee Paralysis Virus-(ABPV). As in the previous case, it is a virus that affects the honey bee and that can also be transmitted using the varroa mite as a transmission vector to predatory species such as *V. velutina* [45]. Recently, a study carried out in northern Spain [47] has detected for the first time ectoparasitic mites of the genus *Varroa* attached to the lateral–ventral part of the abdomen of *V. velutina* specimens. In this study, a prevalence rate of the *Varroa* parasite in the *V. velutina* of 1.75% has been described. Despite this suggestion, there is no study in which the presence of *Varroa* has been detected in the life cycle of *V. velutina.* Such detection would confirm the relationship between the two species. However, this could still suggest a risk in the dispersion of mites that can transmit bee viruses. On the contrary, *Varroa* individuals (probably *V. destructor*, although initially published as *V. jacobsoni*) were detected in *V. vulgaris* larvae from Poland [48]. This finding could make this mite a pathogen affecting this larval stage of this wasp species. There are studies that also demonstrate the role of *Varroa* in the dynamics of hymenopteran pathogens in insect communities. Specifically, they reveal how the hidden effects of the *Varroa* mite can, by spillover, transform the composition of the pathogens in the interacting species [49]. Thus, the arrival of the *V. destructor* mite in Hawaiian populations of *Apis* modified the diversity of the DWV virus in the invasive population of the *Vespula* wasp, with DWV type A being the predominant variant [49].

There are transmission vectors other than *Varroa*. As Vespid species disperse to areas other than their native origins, the detection of new species linked to viral diseases is more evident. *Pneumolaelaps niutirani* (*P. niutirani*) is a species of mite that has recently been discovered in New Zealand [50], hence the name *niutirani*, which derives from the Maori “Niu Tīrani”, which means “New Zealand”. It is a species of mite identified for the first time in association with the nests of the wasp *Vespula germanica*. It has also been discovered in other *Vespula* species (*V. germanica* and *V. vulgaris*) in both New Zealander and Belgian individuals [50,51]. The study of pathogens associated with *Vespula* wasp nests has made it possible to detect viruses such as Moku and DWV in the *P. niutirani* mite [51]. This association between *P. niutirani* with specific pathogens could lead to invasive wasp infections. In addition, this mite can be transported by other species of Hymenoptera such as bumblebees and honey and solitary bees, making it a possible vector of viral transmission in the interaction between different species of Hymenoptera [50,51]. This finding creates new opportunities for the development of a biological control agent. However, this goal would require a better understanding of their feeding, biology, and the mechanism of transmission of pathogens in wasps.

### 4.2. Other Transmission Vectors

There are routes of transmission different to the mites such as the *Nosema* microsporidium [52]. Several species of the genus *Nosema* cause the disease known as nosemosis. *Nosema apis* (*N. apis*), *Nosema ceranae* (*N. ceranae*), *Nosema neumanni* (*N. neumanni*) are the most studied and have been found related to the honey bees [53]. In addition to causing nosemosis, this microsporidium (*N. apis*) could transmit and increase the spread of some viruses such as Black Queen Cell Virus (BQCV) [19]. Although it mainly affects honey bee pupae, this virus can infect the midgut of adult honey bees, probably transmitted by ingestion of *Nosema* microsporidia, with a correlation between peak infections of both pathogens [54]. As is the case with *Varroa*, *Nosema* can be transmitted to predatory Vespids such as *V. crabro* or *V. velutina* [19,22,23], *Polybia scutellaris* [55] and even to pest species for honey bees, such as the small hive beetle (*Aethina tumida*) [56]. Although both pathogens have been identified separately, further studies are needed to determine their possible infection and/or co-infection in Vespids.

## 5. The Incidence of Bee Viruses on *Vespa*, *Vespula*, and *Polistes*

Honey bee viruses have been demonstrated to have an ability to infect other hosts such as the Vespidae species. One of the most defended hypotheses that explains the presence of viruses in Vespids is the one that supports this infection as being caused by a predator–prey interaction [20,21,25,45,57]. The increase in host range could determine the long-term evolution and virulence of pathogens, as well as the emergence and spread of new diseases [58]. However, there are studies that currently provide information on infection due to other vector-type transmission factors.

The scientific literature focused on viruses found in different species of the family Vespidae is scarce. However, further knowledge is essential because social insects with large families play an important role in ecosystems. Moreover, some of them, such as *V. velutina*, have an intense predatory effect on bee colonies. This close relationship could favor the exchange of pathogens between the species and enables *V. velutina* to act as a vector for transmitting pathogens to other bee populations. The main investigations regarding the presence of the most common honey bee viruses in Vespid species are presented below. This study was carried out considering a total of 117 documents found in the time frame of 1968–2021. The search focused especially on those documents that identified bee viruses in the Vespid genera with the greatest capacity to invade new ecosystems. This search allowed for a characterization of bee viruses in the genera *Vespa*, *Vespula* as honey bee predators and *Polistes* as species that share ecosystems with bee viruses hosts. 

A total of 22 viruses (Acute Bee Paralysis Virus (ABPV), Acypi-like Virus, Aphid Lethal Paralysis Virus (ALPV), *Apis mellifera* Filamentous Virus (AmFV), recently identified Macula-like Virus (BeeMLV), Black Queen Cell Virus (BQCV), Chronic Bee Paralysis Virus (CBPV), Deformed Wing Virus (DWV), Israeli Acute Paralysis Virus (IAPV), Ifla-like virus, Kashmir bee virus (KBV), La Jolla virus, Lake Sinai Virus (LSV), Luteo-like virus 1, Menton virus, Moku, Mott mill virus, Nora-like virus, Partiti-like virus, Permutotetra-like virus, and Triato-like Virus) have been found in Vespids. These viruses are summarized in Table 1, Table 2 and Table 3. 

The tables specify where the replication was detected. The tables refer to the following 15 species. Six belonged to the genus *Vespa* (*V. velutina*, *V. crabro*, *V. bicolor*, *V. affinis*, *Vespa* sp., and *V. orientalis*), four to the genus *Vespula* (*V. vulgaris*, *V. germanica*, *V. pensylvanica*, and *Vespula* sp.), and five to the genus *Polistes* (*P. chinensis*, *P. humilis*, *P. rothneyi*, *P. fuscatus*, and *P. metricus*).

In addition, this review highlighted the description of bee viruses in *V. velutina*, as it is an invasive alien species whose rapid and impactful introduction could be altering the composition of pathogens in invaded ecosystems (Figure 1).

### 5.1. Deformed Wing Virus (DWV)

One of the most main viruses associated with the collapse of the colony is DWV [72]. This can be observed in the fact that all studies have demonstrated the presence of this virus in the different genera investigated. It is a positive-stranded RNA virus [72,73], which causes a deformation of the wings of honey bees. The disease is characterized by morphological abnormalities that include malformed appendages, a reduced abdomen, and erroneous coloration [74]. Being one of the most widespread viruses in *Apis*, the study of species that interact with this species, such as Vespids, is also a highly pursued objective. The transmission of DWV in honey bees is usually associated with the mite *V. destructor* [72,74,75,76]. However, it has also been associated with secondary infections by other pathogens [77]. In addition, a horizontal transmission can exist from the pollen in flowers, pollen loads and other bee products to insects [17,29,65,76,78]. High levels of infection in the hive may be the cause of DWV transmission to different predators belonging to the Vespidae family. As in honey bees, the main visible symptom in Vespids has been the presence of deformed wings. This symptom, such as the predation of honey bees, has been evaluated to genetically characterize the existing variants of DWV in Vespids. There are three DWV genetic variants defined as type A, B, and C [25,70,79], with type A being the most virulent [79]. In addition to these three variants, a subtype of this virus known as the Kakugo virus (KV) has been studied, which usually appears with variant A [60]. Table 1 and Table 2 show its presence in *Vespa*, *Vespula*, and *Polistes* individuals. In the case of *V. crabro* [25,60], the presence of the replicating virus in the abdomen and thorax of asymptomatic and symptomatic individuals has been found [25,62]. In *V. velutina*, DWV variants A, B and C have been detected [20,21,45,60]. The presence of variants A and B has also been detected in adult individuals and pupae of other species of the *Vespa* genus, such as *V. bicolor*, *V. affinis*, and *Vespa* sp. [60]. Power et al. [63] have identified this virus in the recently introduced *V. orientalis* in Italy. The DWV has been demonstrated to be present in more than 64 species of insects, including all the species of the genera *Vespula* and *Polistes* (with the exception of *P. humilis* and *P. rothneyi*) treated in this document [65,66,69,71]. *Polistes* spp. are common social wasps that hunt caterpillar prey and feed on nectar by occasionally visiting flowers [80]. It would then be a horizontal transmission that could explain the presence of DWV in this species. Nonetheless, studies exist which associate DWV infection in *Polistes* spp. with the presence of *Varroa*, since when *Varroa* is absent in the ecosystems shared with *Polistes*, DWV has not been detected either. This suggests that *Varroa* could promote increased horizontal transmission within a honey bee colony, therefore increasing transmission in the environment [80].

The replication of this virus has also been studied in the various species of Vespids. In *V. velutina,* the replicative form has been detected by Mazzei et al. [21] in samples collected in Italy and by Dalmon et al. [45] in France, demonstrating active viral replication. Of note is the finding of replicative forms in *V. germanica* from New Zealand and Europe [66,69], *V. vulgaris* in samples from New Zealand [66] and in *Vespula* sp. from the USA [71].

### 5.2. Black Queen Cell Virus (BQCV)

Another virus that has been extensively and frequently studied in Vespids due to its high incidence in honey bees is the BQCV [19,45,60,63,64,65,71]. This virus was also found in all the genera (Table 1 and Table 2). The action of the BQCV in the honey bees causes the death of queens in the larval or prepupal stages. The clinical symptomatology implies a dark coloration accompanied by decomposition, presenting the queen cells’ black spots on their walls. If the queen were to complete her development, she would darken, showing a dark-brown coloration [81]. This virus does not affect adult worker bees, because although they contain the virus, they do not show alteration or symptoms [82]. The most studied transmission mechanism in honey bees is due to contagion during larval feeding [82]. However, it is noteworthy that its presence correlates with the presence of *N. apis*, as mentioned previously [19,54,83]. Like DWV, the spillover effect of this virus is evident, causing infection in several species, including those of Vespidae [58]. BQCV has been studied globally, through its identification in Vespid species, where a high prevalence (from 2.2% to 67%) has been recorded [45]. In this sense, a pronounced genetic diversity of this virus is suggested from the phylogenetic point of view [60]. This virus has also been observed in *V. bicolor*, *V. crabro,* and *V. velutina* [19,45,60]. In addition to being found in a co-infection with *N. apis*, this virus has been observed in co-infection with the Kashmir Bee Virus (KBV) in *V. velutina*, where the absence of vertical transmission is demonstrated [19].

The replicative form has also been found in the gut and muscle samples of this species, as well as in honey bees collected at the same time, suggesting transmission by predator–prey [45]. As for other Vespids species, it has been found in *V. vulgaris*, *P. rothneyi*, *P. metricus*, and *Vespula* sp. [7,60,65,71]. It is of special attention in the study by Levitt et al. [71], since this virus has been found in several insects and arachnid species. This work mentions the infection potential of this virus, even though replication was not studied.

### 5.3. Viruses That Cause Paralysis

Some viruses cause paralysis in some of the stages of *A. mellifera*; these are the Slow Bee Paralysis Virus (SBPV or SPV), Aphid Lethal Paralysis Virus (ALPV), Israeli Acute Paralysis Virus (IAPV), Chronic Bee Paralysis Virus (CBPV), and Acute Bee Paralysis Virus (ABPV) (Table 1). SBPV is an inflavirus with phylogenetic proximity to the Moku virus [61], which was discovered during the bee virus X propagation experiments in England in 1974 [84]. It affects adult honey bees by paralyzing their anterior legs [84]. This virus affects both the larvae and pupae of honey bee, although symptoms were not reported [85]. Its transmission has been related to the *V. destructor* mite and occurs especially during the mites’ feeding activities [86]. Although it has been studied in Vespids (*V. velutina*) [19], it has not been detected in any specimen.

The ALPV, a pathogenic aphid RNA virus, was first identified and classified in South Africa; based on its biophysical and biochemical properties, it was classified as a picorna-like virus [87]. It is currently included in the Dicistroviridae family [88]. Several subgroups with the ability to infect bees have been determined, such as the ALPV-Am strain identified in Spain and in the United States [89], and the ALPV-Ac strain similar to ALPV-Vv isolated in China [59]. In the genera studied, it has only been detected in *Vespa*, and exactly in individuals of *V. velutina*. Major studies demonstrated that this virus has been identified in *V. velutina* hornets in China [59], France [45], and Italy [20]. They demonstrated that *V. velutina* can be an important viral reservoir between *V. velutina* and honey bees, resulting in potential impacts on hive health [59].

IAPV is a virus recently incorporated into the Dicistroviridae family. It has been discovered as a consequence of the spread of the virus in the pupae of white-eyed honey bees [90]. IAPV is a virus that displays high genetic diversity [60] and it is well established in *Apis* colonies, hence its multiple forms of transmission, both horizontal and vertical [91]. *Varroa* can be a vector for the transmission of this virus and its association also promotes the replication of IAPV, causing more damaging effects on honey bee colonies [24,46]. This virus has also been detected in sperm through artificial insemination, causing infections in the tissues of the honey bee queen [91]. This has confirmed the possibility of sexual transmission. Like DWV, its establishment in *Apis* populations allows a spillover effect towards other species that interact with *Apis*, in both predators and non-predators. Therefore, IAPV uses a wide variety of carriers and additional hosts [46,65]. In Vespids, the literature suggests that IAPV infection is generally related to bee ingestion [19,46]. Thus, this virus has been detected in wasps whenever the virus is also present in nearby bee populations [65]. As main results, IAPV has been detected in *V. velutina* [30,46], *V. bicolor*, *V. affinis,* and *Vespa* sp. [30] (Table 1), *V. vulgaris* [65], *V. germanica* in its replicative form [69] and *P. rothneyi* [60] (Table 3). It is important to add that this virus has been detected in pupae samples of some Vespids such as *V. bicolor*, *P. rothneyi*, *V. affinis*, and *V. velutina* [60]. For infection in pupae, feeding by adult individuals suggests the possibility of virus transmission. However, Yang et al. [60] suggests that it would be necessary to go deeper into the knowledge of infection in pupae, since it could be a more susceptible stage for the replication of this virus since they come from larvae fed on large quantities of masticated bees.

Chronic bee paralysis virus (CBPV) was isolated in *Apis* [92] and described in France as “maladie noire”; it causes a Type 2 syndrome [73,92,93] characterized with clusters of flightless trembling and crawling bees. In addition, some black individuals standing at the entrance of the hive are described in infected hives [92,94]. Furthermore, this virus can cause another version of infection known as Type 1 syndrome, first described in the UK. This is characterized by trembling and crawling bees and by less frequent or null black individuals and occurs at any time of the year [77,93]. These syndromes correspond to those described by Ribière et al. [94], who consider that it is composed mainly of two segments, RNA 1 and RNA 2. One of the syndromes causes the appearance of dysentery that cannot be excreted and is known as “Bloated abdomen”; the other syndrome is known as “black thieves” or “hairless black syndrome”. CBPV causes a very contagious infection with high multiplication levels causing very significant losses in bees that can be aggravated when accompanied by nutritional deficiencies or bad weather conditions in the summer [94,95]. The high replication of this virus promotes its spread to other species such as ants [96] but also Vespids [45]. In Europe, CBPV has been studied in *V. velutina* [19,23] for its high contagion when predating honey bees. However, only the study carried out by Chauzat et al. [23] demonstrated the presence in *V. velutina* from France. The presence of the virus, in coinfection with SBV (Sacbrood virus) and BQCV viruses, in diseased larvae with pathogens such as *A. woodi* and *Nosema* spp., was highlighted in the latest study. Symptomatology involved the non-conversion of pupae and a change in larvae color from pearly white to pale yellow and turning dark brown [23].

Parallel to the discovery of this latter virus (CBPV), the acute bee paralysis virus (ABPV) was discovered [77,93,94,97]. ABPV causes tremors and paralysis in a few days after infection; however, individuals die earlier than with chronic paralysis. High levels of the virus, at around 100 particles of ABPV, generate symptoms of paralysis, abnormal wings and body tremor in 2–4 days and the death in 1 or 2 days more [93]. Regarding the Vespid species, to date no symptoms have been detected. ABPV has been studied in Vespids, such as *V. velutina* [19,45,46,60] or *V. germanica* [69]. Regarding *V. velutina*, infection with this virus was detected in muscles and/or cuticle, legs, and mandibles, as well as in the guts of the hornet [45]. The presence of the virus in the body of *V. velutina* could be related to the infection cycle; however, further experimentation is necessary to characterize the virus and its virulence in the species [45]. In *V. germanica*, the replicative form has been detected [69], as shown in Table 2. It is important to note that the sequences analyzed in *V. velutina* [45] demonstrated 99% identity with the acute bee paralysis virus (ABPV) isolated Hungary 1 (GenBank AF486072.2), but with differences with other ABPV sequences obtained from bees of the same experiment. Furthermore, phylogenetic analysis in *V. germanica* demonstrated that the ABPV sequences were more closely related to each other than to the global GenBank sequences. These different sequences to others could support the hypothesis that Vespid species could be, in addition to carriers, hosts for virus infection.

ABPV is frequently studied with the Kashmir bee virus (KBV) and Israeli acute bee paralysis virus (IAPV), being a part of the AKI complex. These viruses are closely related virus species from the Dicistroviridae family [97,98]. ”AKI” is a complex formed by three viruses (acute bee paralysis virus ABPV, Kashmir bee virus KBV, Israeli acute bee paralysis IAPV) that have a worldwide distribution [45]. The presence of this complex is related to the high loss of bee colonies, especially when the colonies are infested with the parasitic mite *V. destructor* [97]. KBV is an RNA virus belonging to the Dicistroviridae family [99] and is one of the most virulent viruses that affect *Apis* [81,100]. This virus has been detected in the feces of both worker and queen bees and feeding is the most probable cause of transmission [65]. In addition, a vertical transmission has been detected; this would be in the case of the infection of bee queens that could transmit the virus through the eggs (*transovarian* transmission) [97,100,101]. The *V. destructor* could be involved in the initiation of KBV replication, which can also transmit KBV between adult bees and pupae [85,99,102]. It is genetically related to the acute bee paralysis virus (ABPV) and both can co-infect the same hive and the same honey bee. Some factors, such as stress in the hive, could propitiate to increased infection of this virus in the hive [19]. KBV has been detected in the genera *Vespa* and *Vespula*. In the case of *V. velutina* [19,46], an active viral replication in individuals has been demonstrated even at the end of the season. It is worth mentioning that the virus has been detected in parts other than the gut, such as in the brain of the same species [45]. This fact and the finding of replication in this species indicate that this virus could be adapted to *V. velutina* and, therefore, it is not specific to the honey bee. This virus has also been observed in other Vespids of the *Vespula* genus such as *V. vulgaris* [41,51,66,67] and *V. germanica* [51,66,69]. In both species, the replicative form has been detected, as well as coinfection with DWV. The replication of this virus in Vespid species and the high viral load detected in the sampled specimens could indicate that KBV is a potential pathogen of these species. Coinfection with other viruses also opens a range of different hypotheses about the infection mechanisms of this virus in Vespids, such as the weakening of the host defense mechanisms [19].

### 5.4. Sacbrood Virus (SBV)

The Sacbrood virus or Saciform Brood Virus (SBV) is also an RNA virus with a picornavirus in the genus Iflavirus and belongs to the family Iflaviridae [100,103,104]. SBV can present causes failure to pupate and subsequent death in *A. mellifera,* affecting both larvae and adult honey bees [103]. SBV in the early stages can result in the larvae dying shortly after being capped and before pupation. Therefore, the larva is unable to detach from the last larval cuticle and does not reach the operculation stage. The following days, eccidial fluids continue to accumulate between the cuticle and the tissues, causing the characteristic color of this infection and forming the sac [100]. The spread of this virus has been detected in both *Apis* predator and non-*Apis* predator hymenopteran [65,71]. A high prevalence has been also detected in studies that have tested this virus in species other than honey bees [45,60]. Regarding the main Vespid species in the literature, this virus has been observed in *V. velutina* [45,60]. Other species in which this virus has been detected were *V. vulgaris* [64,65], *P. metricus*, North American native wasp [65] and *Vespula* sp. [71] (Table 2 and Table 3). In the case of *V. velutina*, it has been detected in the part of the gut. This could lead to the suspicion that the transmission of this species occurs by preying on honey bees. However, as mentioned, it is a virus with a high prevalence, which has even been detected in pollen pellets and in non-predatory species of honey bee. In addition, finding this species in a stage such as the pupa of *V. vulgaris* (without symptoms of infection), it can be thought that it is a well-established virus and a pathogen for a high diversity of pollinating hosts.

### 5.5. Moku Virus (MV)

The inflavirus Moku virus (MV) was first discovered in the predatory social wasp *Vespula pensylvanica*, probably its native host, on Big Island, Hawaii [70]. It is an RNA virus, genetically different from any other virus at the nucleotide level, hence its nomenclature as “Moku”, which means island in Hawaiian. A close relationship has been found between this virus and the slow bee paralysis virus in studies on Belgian individuals of *V. velutina* [61]. However, no symptoms have been described for this virus. The Moku virus has also been detected in honey bee and *Varroa* samples. Although it was detected for the first time in *V. pensylvanica* [42,70,105], it has also been identified in other Vespids such *V. crabro* in UK [57] or *V. velutina* in Belgium [61] and France [45]. It has also been detected in *P. chinensis* and *P. humilis*, *V. germanica*, and *V. vulgaris* [66,68]. It is worth mentioning that in the specific case of *V. velutina*, as it is a predator that exerts great pressure on honey bees, a transmission in the direction of bees to Vespids could be suggested. Furthermore, the sequencing data and the higher abundance of this virus in Vespids versus honey bees could reinforce this suggestion and pose a future risk in the transmission of the virus from hornets to honey bees during predation events in hives [57,61,70]. However, it would be necessary to carry out more studies on the transmission of this virus that could demonstrate it. This virus has also been detected in co-occurrence with other pathogens; thus, in individuals of *V. vulgaris* from both New Zealand and Belgium, it has been detected in co-infection with high viral diversity. The majority belong to the Picorna family, but other families such as Luteo-Sobemo, Partiti and Tombus were also detected [68]. Of note is the detection of *Vespula vulgaris* Luteo-like virus 1, whose high presence in all life stages suggests a persistent infection in *V. vulgaris* colonies [68]. It is important to point out that the replicative form has also been detected, as it has been found in individuals of *V. crabro* [25], *V. pensylvanica* [42,57], *P. chinensis*, *V. germanica,* and *V. vulgaris* [66], as shown in Table 2 and Table 3. Although the Moku virus can infect honey bees, it is better established in Vespid species. In fact, it has been proven that in populations of *V. vulgaris*, the viral load of the Moku virus can affect vital traits such as the longevity of the colony [42].

### 5.6. Lake Sinai Virus (LSV)

LSV, belonging to the genus Sinaivirus, was discovered in Western honey bee (*A. mellifera*) samples near Lake Sinai, South Dakota, USA [106]. Initially, only two variants known as LSV1 and LSV2 were discovered that can infect continuously over time. Thus, LSV2 could reach maximum peaks in January and April while LSV1 in July [106,107]. These two viruses can give symptoms especially when the bee colony is weakened by other factors such as colony collapse disorder (CCD) [108] or *Nosema* spp. [109]. More variants were later detected and included in this virus—LSV3 [108], LSV-Navarra [108], LSV4, LSV5 and several LSV discovered in Belgium [110,111]. LSV1 and LSV2 are characterized by not presenting obvious symptoms and by their predominance in adults [107]. Contradictorily, the identification of this virus in Vespidae species has only been referred to its presence in a *V. bicolor* pupa from China [60]. In this study, an occurrence has also been observed with other viruses such as *Apis mellifera* Filamentous Virus (AmFV), IAPV, DWV, CSBV, and BQCV. However, given that no virus replication was found in these samples, the appearance of the virus is probably due to contamination by feeding.

### 5.7. Apis Mellifera Filamentous Virus (AmFV)

AmFV was originally described as a rickettsial disease of honey bees (*A. mellifera*) [112]; however, it was later characterized as a large, enveloped DNA virus [107,112]. One of its main diagnostic features is that it renders the hemolymph of adult bees milky white due to cell degradation and the large number of rod-shaped virions present, when examined by electron microscopy [73,107,112,113]. Despite being a widely distributed virus, studies on other possible hosts, such as Vespids, are scarce. In fact, only the study by Yang et al. [60] has identified AmFv in hornets from China, in *V. bicolor* pupae and in both pupae and adults in *P. rothneyi* [60]. In these detections, no symptoms have been detected. Coexistence with other pathogens such as IAPV, DWV, CSBV, BQCV, and LSV has also been detected in *V. bicolor*, while in *P. rothneyi*, AmFV has been detected together with BQCV or together with IAPV [60]. In honey bees, it can occur together with *Nosema* [75] as a prerequisite; however, in Vespids this association is not mentioned. The scant information on its transmission in Vespids only explains that its presence is due to consumption by bees, hence its presence in larvae. Furthermore, the fact that there is no evidence yet for Vespid-specific sequences suggests that Vespids are only carriers of this virus.

### 5.8. Bee Viruses Detected Exclusively in V. Velutina

Recently identified Macula-like virus (BeeMLV) is a polyadenylated RNA virus in the family Tymoviridae in the order Tymovirales and close to the order of Maculavirus [75,114]. The presence of BeeMLV in bees has been related to *V. destructor*, in which the replicative form of the virus has been detected, suggesting its participation as a possible transmission vector [75,114]. In honey bees, this virus can infect both pupae and adults. However, in Vespids it has only been detected in the gut of adult individuals of *V. velutina* from France; thus, it is suggested that the presence of this virus is probably related to bee predation [45].

It is also worth mentioning the La Jolla virus (LJV) belonging to the Iflaviridae family, discovered for the first time in *Drosophila* and detected in honey bees in Australian samples [115]. This virus was detected for the first time in Italian samples of *V. velutina* [20]. In that study, the presence of La Jolla virus in bee samples was also confirmed, suggesting transmission through environmental contamination. Although no symptoms have been reported in *V. velutina*, melanization followed by rapid death has been observed in *Drosophila* within 3 days after injection of the LJV strain. This contribution has focused on LJV as a good biopesticide candidate for this species [20]. However, in the case of *V. velutina*, a better characterization of this virus would be necessary.

Finally, other viruses associated with the analysis of *V. velutina* from France were also detected, such as the Acypi-like virus, Triato-like virus, Permutotetra-like virus, Partiti-like virus, Ifla-like virus, Nora-like virus, Menton virus, and Mott mill virus [45].

The rapid spread of *V. velutina* in invaded ecosystems, coupled with its ability to carry a wide range of pathogens, is indicative of the potential risk it may pose to the species with which it interacts, such as honey bees. Most of these viruses have been detected as the target of a search for control of this pest species [20,21,22,23]. The adaptation and accommodation of *V. velutina* in the invaded ecosystems brings about the presence of new interactions between viral pathogens and insects. A potential indicator of this event could be represented by the ratio of the replicative virus/non-replicative virus [20]. In *V. velutina*, the presence of replicative forms of several of the viruses mentioned above has already been confirmed. A total of nine viruses have replicated in samples of this hornet (Table 1).

The literature found regarding the distribution of *V. velutina* in areas other than the native corresponds to three countries in Europe (Figure 1a), in which this pathogenic diversity has been detected. France is one of the countries invaded by *V. velutina* that has carried out the largest number of investigations into the bee virus. Figure 1 shows the main bee viruses identified in this country, highlighting the replicative forms of the viruses DWV-B, BQCV, SBV, ABPV, Acypi-like virus, and Triato-like virus that were identified [45]. France is followed by the Italian and Belgian studies, where replication has only been found in Italy (DWV-A; DWV-B; DWV-C, BQCV, KBV, ALPV, and Triato-like virus [19,21]). *V. velutina* is native to Southeast Asia, being common in southern China [14,15]. From these areas, it has been able to invade along two routes, through northeast Asia and Europe. In northeast Asia, it spread through South Korea to Japan [15], where it has been introduced as an invasive species (Figure 1b). The main studies on bee viruses in Asia have been carried out with individuals from China. From these studies, the replicative forms for the APV and ALPV viruses were found [46,60]. These studies suggest and reinforce the existence of colonization in these environments through the establishment of a balance between pathogenicity factors and *V. velutina*.

## 6. High Rate of Evolution of Bee Viruses as a Future Risk in Invaded Ecosystems

Bee viruses represent a threat to the loss of honey bee populations. For this reason, special attention should be paid to their propensity to jump between host species and, therefore, threaten other populations of wild pollinators with important ecological and economic roles [116]. This spread also affects the introduction of non-native species, such as Vespids. The research discussed in this review demonstrates that there is a virus transmission from honeybees to Vespids, and that virus replication does occur on certain occasions. This highlights the need to reformulate the specificity of bee viruses for honey bees and to deepen the scarce knowledge on their spread and infection of new hosts.

In contrast, there are no studies demonstrating reverse directionality, that is, from the Vespids to honey bees. However, the literature highlights the spillover of bee viruses into ecosystems invaded by invasive Vespids. This poses a risk for the spread of bee viruses and considers a gap regarding the future of bee virus control in apiaries.

The viruses presented above are often single-stranded RNA positive (+ss) viruses. They have a high replication rate and high error rates, resulting in them being genetically very heterogeneous. Some of the bee viruses discussed in this review, such as DWV, have been sequenced and structurally characterized. DWV can present three different variants with different virulence (variant types A, B, and C). Some molecular studies estimate a mean evolutionary rate of 1.346 × 10^−3^ substitutions/site/year [117], allowing the rapid adaptation to new host environments, with recombination representing an additional key source of genetic variation.

The main infections caused are usually transmitted by the invasive mite *V. destructor*. This may have favored the evolution of the virulence of the bee viruses it transmits and driven the emergence of viruses in other species in areas where it has been introduced. In parallel, the occurrence of these viruses in Vespid species has revealed more specific replication dynamics in Vespids. However, the lack of knowledge about this type of occurrence in Vespids may suggest that there is a future risk of changes in infection dynamics due to the high rate of evolution of these viruses. This review therefore highlights the need to study bee viruses in invasive species from the time of their early introduction.

## 7. Conclusions

An increasing number of shared viral genomes have been found between the family Apidae and the family Vespidae. These viruses are RNA viruses with a high rate of evolution, allowing a high frequency of virus exchange within and between the families of Hymenoptera. This could lead to the reformulation of the definition of species-specific viruses and specifically for the bee. Knowledge about bee viruses in Vespid species is scarce. Most of the investigations describe the presence and/or replication of the bee virus in Vespids. However, very little is known about the transmission mechanisms of these viruses. *V. velutina*, which has an increasing spatial distribution outside its native range, is having a major impact on beekeeping. This has promoted a growing number of investigations on bee viruses in this species. No less impactful are other invasive species that prey on bees or share their habitat, whose introduction has changed the dynamics of virus replication. These species could be a potential factor in the transmission of bee viruses, affecting the health and strength of the colony. The information collected in this review on honey bee viruses in invasive Vespids is relevant to promote the further research on honey bee virus transmission in Vespids and improve the management of infectious diseases caused by bee viruses in apiaries.

## Figures and Tables

**Figure 1 insects-14-00006-f001:**
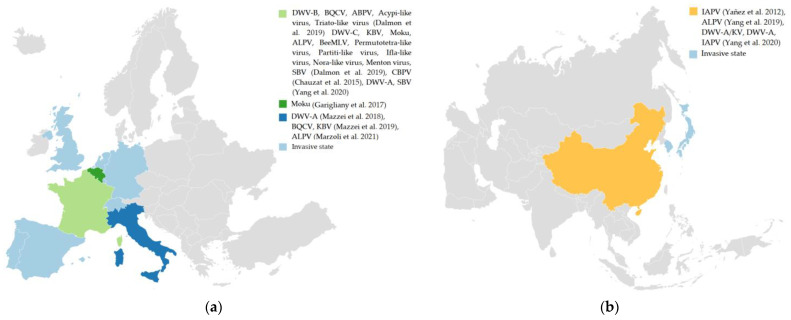
Studies on bee viruses of *V. velutina* in the invasive area in Europe; (**a**) and in its native area; (**b**) In addition, figures also show its distribution as an invasive species in light blue (Invasive state). Created with Datawrapper. The references correspond to the following call-out numbers: Dalmon et al. [45], Chauzat et al. [23], Yang et al. [59,60], Garigliany et al. [61], Mazzei et al. [19,21], Marzoli et al. [20] and Yañez et al. [46]. Created with Datawrapper.

**Table 1 insects-14-00006-t001:** Occurrence of viruses in *Vespa*.

*Vespa* Species	Viruses	Occurrence
*V. velutina*	ABPV	R [45]
	Acypi-like virus	
	ALPV	X [45], R [20,59]
	BQCV	R [19,45]
	DWV-A	X [60], R [21]
	CBPV	X [23]
	DWV-B	R [45]
	DWV-C	X [45]
	IAPV	X [60], R [46]
	KBV	X [45], R [19]
	La Jolla virus	X [20]
	Moku	X [45,61]
	Ifla-like virus	X [45]
	BeeMLV	
	Menton virus	
	Mott mill virus	
	Nora-like virus	
	Partiti-like virus	
	Permutotetra-like virus	
	SBV	X [60], R [45]
	Triato-like virus	R [45]
*V. crabro*	DWV-A	R [25,62]
	Moku	R [57]
	BQCV	X [60]
	DWV-A	
*V. bicolor*	AmFV	X [60]
	DWV-A	
	DWV-A/KV	
	IAPV	
	LSV	
*V. orientalis*	ABPV	X [63]
	BQCV	
	DWV-A	
	KBV	
*V. affinis*	DWV-A	X [60]
	IAPV	

X indicate occurrence and R indicate that the virus is replicative.

**Table 2 insects-14-00006-t002:** Occurrence of viruses in *Vespula*.

*Vespula* Species	Viruses	Occurrence
*V. vulgaris*	BQCV	X [64,65]
	DWV	X [65], R [66]
	IAPV	X [65]
	KBV	X [67], R [41,66]
	Luteo-like virus 1	X [68]
	Moku	X [67,68], R [66]
	SBV	X [64,65]
*V. germanica*	ABPV	R [69]
	DWV	R [66,69]
	IAPV	R [69]
	KBV	X [66], R [69]
	Moku	R [66]
*V. pensylvanica*	Moku	R [42,70]
*Vespula* sp.	DWV	R [71]
	KBV	X [51]
	Moku	
	BQCV	X [71]
	SBV	

X indicate occurrence and R indicate that the virus is replicative.

**Table 3 insects-14-00006-t003:** Occurrence of viruses in *Polistes*.

*Polistes* Species	Viruses	Occurrence
*P. chinensis*	Moku	R [66]
	DWV	X [66]
*P. humilis*	Moku	R [66]
*P. rothneyi*	AmFV	X [60]
	BQCV	
	IAPV	
	LSV	
*P. fuscatus*	DWV	X [65]
*P. metricus*	BQCV	X [65]
	DWV	
	SBV	

X indicate occurrence and R indicate that the virus is replicative.

## Data Availability

Not applicable.

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
