# Peer review of "Emerging Risk of Cross-Species Transmission of Honey Bee Viruses in the Presence of Invasive Vespid Species"

_insects, 2022, doi:10.3390/insects14010006_

Round 1
Reviewer 1 Report
The work seeks to bridge an important gap in our knowledge between social wasps and the honey bees (a relatively close evolutionary relative) by bringing together and synthesizing a variety of disconnected works. Such a review is well-needed but this attempt is far from ready for publication. It will require major revision to the language, content, figures, and citations. The English in this article is not consistently clear and needs substantial revision throughout to avoid detracting from the quality of the overall work. I have provided examples of several forms of typos and writing issues systemic in this article without pointing out each issue directly. To refine this publication, the authors should go through each line of this text and check the English for clarity, grammar, and correct syntax. Further, the citations and figures need revision as well. The figure representing the genetic bottleneck seen in introduced species doesn’t seem clear or necessary for the manuscript. Several statements are in need of citation or updated citations. References to “V. jacobsoni” are out of step with current research showing that the authors have not remained up to date on changes in the parasitic mite landscape over the last 2 decades. Further, several sections are vague. A successful review provides detailed, well-synthesized information such that someone reading it understands the most current and up-to-date state of research and where to find more information. Several sections of this review use non-specific language and/or lack pertinent details. The skeleton of this review is solid but several elements of how it’s presented need attention for a second submission to be considered a substantial improvement.

Author Response
The work seeks to bridge an important gap in our knowledge between social wasps and the honey bees (a relatively close evolutionary relative) by bringing together and synthesizing a variety of disconnected works. Such a review is well-needed but this attempt is far from ready for publication. It will require major revision to the language, content, figures, and citations. The English in this article is not consistently clear and needs substantial revision throughout to avoid detracting from the quality of the overall work. I have provided examples of several forms of typos and writing issues systemic in this article without pointing out each issue directly. To refine this publication, the authors should go through each line of this text and check the English for clarity, grammar, and correct syntax. Further, the citations and figures need revision as well. The figure representing the genetic bottleneck seen in introduced species doesn’t seem clear or necessary for the manuscript. Several statements are in need of citation or updated citations. References to “V. jacobsoni” are out of step with current research showing that the authors have not remained up to date on changes in the parasitic mite landscape over the last 2 decades. Further, several sections are vague. A successful review provides detailed, well- synthesized information such that someone reading it understands the most current and up-to-date state of research and where to find more information. Several sections of this review use non-specific language and/or lack pertinent details. The skeleton of this review is solid but several elements of how it’s presented need attention for a second submission to be considered a substantial improvement.
We would like to thank the reviewer for the very detailed corrections he has made to the submitted revision. Each of these comments has been modified to substantially improve the work and each is responded to below. The language has also been reviewed in detail.
Line 8. Key premise is that viruses cause great losses in honey bee populations. Please qualify this statement with actual numbers. Provide a citation for these numbers when this premise is referenced in the text.
The sentence has been changed to the following:
Currently, bee viruses are one of the threats to honey bee populations.
Line 9 “causes an impact” is clunky and unclear language. What kind of impact (i.e. negative or positive). You explain how these impacts occur in the next line.
The authors appreciate this observation and have clarified that this is a negative impact.
Line 14 – remove the word “these” before “viruses” and the sentence will read more clearly.
Done
Line 14-16 – what you mean by “suggests changes in the balance between pathogenicity and host factors, weakening the defense strategies of native species” is unclear. The simple summary should be clear and easy to read. Please revise your language to provide clarity.
The authors change the sentence for a better comprehension: “The presence of viruses in species of vespids that interact with honey bees could represent an emerging risk in the transmission of pathogens, weakening the defense strategies of native species.”
Line 24 – “Vespid” is a proper noun and should be capitalized throughout.
Done
Line 25 – The role of Varroa in virus transmission is still very murky. We know that they can transmit several viruses but there are several more where the answer is unclear. And we still do not know for sure whether the viruses use Varroa as a host in addition to the bees. Please revise this line to reflect that there is more clarity about the role of Varroa in virus transmission than in Vespids but not necessarily clarity throughout on the subject of Varroa.
The text has been clarified: “ The role of some mites such as Varroa in the transmission of honey bee viruses is clearer than in the case of Vespidae. This type of transmission by vectors has not yet been clarified in Vespidae, with interspecific relationships being the main hypotheses accepted for the transmission of bee viruses.”
Line 26 Interspecific relationships are also argued between Varroa and honey bees making it difficult to know what distinction you’re attempting to draw here in comparing theories on Varroa and Vespids.
In full agreement, the text has been changed to the following:
“The role of some mites such as Varroa in the transmission of honey bee viruses is clearer than in the case of Vespidae. This type of transmission by vectors has not yet been clarified in Vespidae, with interspecific relationships being the main hypotheses accepted for the transmission of bee viruses.”
Line 27 – typo: should be “viruses” not “virus”
Done
Line 44 – In a single paragraph you’ve referred to them as “exotic, invasive, non-indigenous”. Your writing will be clearer if you choose on and maintain consistency.
The authors are aware of the use of these terms, however, in the case of the term exotic, it means foreign prior to its introduction. In the rest it is due to a simple reason of not repeating the same word in a very continuous way. However, we understand the non-uniformity so in the case of non-indiggenous it will be changed to invasive or non-native.
Figure 1 is not clear in helping to highlight your point. The image which purports to show a bottleneck doesn’t clearly illustrate the narrowing diversity observed in these contexts. The hymenopteran on the right side of the image seems superfluous as do the arrows emanating from the large blue circle.
Additionally, the concept of a genetic bottleneck is fairly well-known and this illustration doesn’t represent it in a way that justifies its presence.
The authors have decided to eliminate Figure 1, since it has been considered that the explanation in the text is sufficient.
Line 51-55 – It won’t be clear to readers how both can be true (that the likelihood of pathogens being transported by invasive species is low and the expansion of invasive species plays an important role in global spread of pathogens). Please explain further such that these sentences don’t immediately present as confusing.
The authors refer to the increased spread of invasive species that could be contributing to changes in ecosystems, including the increase in pathogens and infectious diseases.
The text has been modified for better understanding
Line 55 – “especially the greater connection and globalization” language should be revised for clarity. Line 65 – typo: “Vespula, Vespa, and Dolichovespula are a pest” should be “are pests”
Done
Line 69-71 – typo: should be “introduced”. Also, an organisms is not “introduced as an invasive species” but becomes one after presenting damage to the environment, competition for resources with established species, or threat to human health. Thus, the content of the sentence is redundant. Please revise. The sentence further needs a citation for the concerns stated about the three species listed.
The changes suggested by the reviewer have been made.
Line 72 – It is not clear what “Regulation Nº 1143/2014” is. Please provide a proper citation
Done
Line 74 – Which Asian hornet are you referring to as “the Asian Hornet” as several are called by that generic name. Please include the binomial name.
Done
Line 74-75 – the common name for Apis cerana is not “Asian Bees” but instead “the Asiatic honey bee” or the “Eastern Honey Bee”. Please also provide a citation for the premise of this sentence that V. velutina populations are kept in balance in their native range. Additionally, there is a typo. It should be “prey on” not “prey”. I’ve pointed out several styles of typos and grammatical issues in the first two pages. There are too many others for me to continue pointing out all of them. Please use the information provided herein to go through this entire manuscript line by line to find the remaining typos and grammatical issues. From this point onward I will provide less and less detail on each individual typo and rely on you scour this paper and find the rest.
Reference to this statement has been provided and typographical errors have been corrected.
Line 79 – typo. And the name “honey bee” is properly written in scientific words as two words. Please correct throughout.
Done
Line 78-80 – citation needed
Done
Line 82 – insects are small animals. You have already said that they consume insects so they consume “other small animals”
Done
Line 83-84 – revise for clarity of thought. It’s a risk factor for what? Why is greater connection between nodes a problem?
The authors wanted to highlight the contact it has with multiple ecosystem components by taking advantage of a wide diversity of resources.
The text has been changed for better understanding.
Line 86-88 – Grammatical revision needed
Done
Line 88-91 - Grammatical revision needed. Citation needed
Done
Line 91-93 - Grammatical revision needed.
Done
Line 97-98 – “Most invasive Vespid species interact extensively when preying on other insects…” Make it clear in this sentence that you mean they are interacting with prey species not each other. The sentence can currently be interpreted either way as written.
In this sentence the authors would like to highlight the interaction between different species of insects as well as flora when visited by vespids.
The text has been modified for a better comprehension: “Most invasive Vespid species interact extensively with other species, when preying on other insects such as honey bees, and during foraging and the search for resources to build nests [17].”
Line 99-100 – citation needed
The quote is defined by reference 17. Therefore, it has been transposed to the end of this paragraph so as not to be too repetitive and to include the entire text.
Line 105-107 – the word “rebalanced” suggests that there was already a balance but these organisms were naïve to each other prior to the introduction so this phrasing is inaccurate.
The problematic word has been changed to ”reconfigured”
Line 113 – typo
Done
Line 118-119-typo
The sentence was changed to “The transmission of pathogens is another consequence of the introduction of these invasive species [20].”
Line 125-126 – “not a predatory species of bees” is a confusing sentence structure. Please revise. Also, please describe how these species may spread viruses if their primary interaction with bees isn’t predation. You detail the categories of interaction in the next sentence but don’t connect it to Polistes.
The sentence was changed to “Although species of the genus Polistes do not prey on bees, the spillover effect of the different bee viruses in ecosystems makes it a carrier and host for these viruses.”
Line 128-129 – citation needed. Also, it is not all invasive social wasps. Please clarify that.
Done
Line 129-130 – What is “the parallel stage of invasion”?
simultaneous phase of the invasion
Line 130-132 – Citations needed (provide examples of some of these literary works you reference in this sentence)
Done
Line 142 – “or” not “and”
Done
Line 145-146 – typo.
The sentence was changed to “The virus is transmitted vertically from the parental generation to the brood through the egg.”
Line 149 – citation needed.
Done
Line 150-151 – citation needed
Done
Line 176 – “Pollen” likely doesn’t need to be included in this sentence as Vespids don’t forage for pollen, just nectar.
Done
Line 190-191 – Revise for clarity. What impact does viral load have on the longevity of the colony?
The paragraph was changed to “The longevity of wasp colonies has also been influenced by the presence of diverse viral loads. This is the case for colonies of the Vespula pensylvanica wasp, whose longevity may be affected by the viral load of the Moku virus [42].
Line 196 – There are many parasitic mite species not “few” as stated here. There are at least as many as there are honey bee species and likely more undiscovered.
The sentence was changed to “Some of the parasitic mite species present in hives can cause serious diseases in honey bees.”
Line 197-198 – Your information here is outdated. Tropilaelaps mercedesae is the most damaging Tropilaelaps species with T. clareae having little relevance outside of the Philippines (Anderson 2007; Genetic and morphological variation of bee-parasitic Tropilaelaps mites (Acari: Laelapidae): new and re-defined species). And Varroa destructor is the Varroa might causing damage in Apis mellifera with little damage caused by V. jacobsoni outside of Oceania (Anderson 2000; Varroa jacobsoni (Acari: Varroidae) is more than one species). In addition, “jacobsoni” is spelled with a single “i”. Citation is needed here as well.
The authors have made the appropriate changes
Line 199-201 – Varroa causes substantial damage directly by feeding on honey bee fat body tissue not just through transmission of viruses.
A change has been made to the sentence as follows: “Among them Varroa, is the causative agent of varroosis one of the most serious diseases in honey bees [24]. In addition, it has the potential to contribute to the worldwide mortality of honey bee colonies through several RNA viral infections [44].”
216-218 – “could interact with the interact with the lifecycle of the host” is unclear language. Interact in what way? What interactions have been observed thus far? And Line 217 – Varroa jacobsoni is currently restricted in its geographic range to Asia and Oceania. This study found V. destructor but prior to this species being identified as separate from V. jacobsoni.
This phrase refers to the fact that being found in the larval stage of V. vulgaris could confer it the role of pathogen. we have changed the sentence to correct the sentence.
On the contrary, Varroa individuals (probably V. destructor, although initially published as V. jacobsoni) were detected in V. vulgaris larvae from Poland [48]. This finding could make this mite a pathogen affecting this larval stage of this wasp species.
Line 221-223 – citation needed
Done
Line 244-246 – Be specific. Which species of Nosema have been shown to interact synergistically with BQCV?
The authors have changed the sentence to the following one, which is less conjectural and clearer than the previous one: In addition to causing nosemosis, this microsporidium (N. apis) could transmit and increase the spread of some viruses such as Black Queen Cell Virus (BQCV) [19].
Line 256-260 – The wording here is confusing. Please revise for clarity.
The paragraph was changed to: “One of the most defended hypotheses that explains the presence of viruses in Vespids is the one that supports this infection as being caused by a predator-prey interaction [20,21,25,45,57]. The increase in host range could determine the long-term evolution and virulence of pathogens, as well as the emergence and spread of new diseases [58]. However, there are studies that currently provide information on infection due to other vector-type transmission factors.”
Lines 264-266 – citation needed. Do we know for sure that V. velutina serves as a transmission vector between bee species?
Although there are no studies on the directionality of virus infection from vespids to bees, the presence of viruses in these, together with the enormous pressure they exert on bees (as is the case of V. velutina), allows us to write the premise that there may be this transmission in this sense.
However, we have changed the text by correcting the typographical errors that did not give clarity to the text.
Table 1 – Viruses with abbreviation that have not been described in the literature should have their full names available in a key or the table description. Some viruses share the same acronym.
Table 1 has been corrected according to the reviewer's suggestion.
“Vespa affinia” is not a recognized species name. It’s Vespa affinis. Revise throughout.
Done
Lines 283-284 – citation needed. Further, your logic here does not follow. Just because a virus is widespread among species doesn’t mean that it will be “one of the most important viruses in the collapse of the hive”.
Authors have clarified the sentence by the following:
One of the most main viruses associated with the collapse of the hive is DWV [72: De Miranda, J. R., & Genersch, E. (2010). Deformed wing virus. Journal of invertebrate pathology, 103, S48-S61. ].
Lines 294-296 – Citation needed! Lines 296-297 – Citation needed!
References have already been provided for the lines mentioned.
Lines 304-306- revise for clarity. Vespa affinis not affinia. And you have another species listed as just Vespa in this list. Please revise.
Done
Line 307 – DWV stands for Deformed Wing Virus, thus “DWV virus” is redundant.
Done
Line 313-316 – citation needed!
Done
Line 327-329 – citation needed
Done
Lines 330-332 – citation needed
Done
Line 340-342 – citation needed!!
Done
Line 343-345 – Please revise for clarity
Done
Line 348 - Wow! Opiliones! Fascinating!
Although this comment is not understood as a correction or suggestion to change, we have chosen to generalize by mentioning only arachnids. In addition, the text derived from the findings discussed is added below.
Levitt, A. L., Singh, R., Cox-Foster, D. L., Rajotte, E., Hoover, K., Ostiguy, N., & Holmes, E. C. (2013). Cross-species transmission of honey bee viruses in associated arthropods. Virus research, 176(1-2), 232-240.
Our study confirmed and extended this observation to include not only new pollinator species, but also other arthropods associated with honey bee apiaries. Viruses surveyed in this study (BQCV, SBV, KBV, DWV, and IAPV) were detected in seven orders belonging to class Insecta (Blattodea, Coleoptera, Dermaptera, Diptera, Hemiptera, Hymenoptera, and Lepidoptera) and Arachnida (orders Araneae and Opiliones) (Table 1).
Line 356 – citation needed
Done
Line 386 – citation needed
Done
Lines 400-402 – citation needed
Done
Lines 408-411 – citation needed and Line 412 – citation needed
Done
Line 438-440 – citation needed
Done
Line 445-446 – citation needed
Done
Line 446-447 – citation needed
Done
Line 454 – citation needed
The sentence has been changed as a premise resulting from the previous results, so that it refers to all the previous references.
Line 461-462 – Your logic hear is unclear. Please revise for clarity. Why is it that infecting adult and juvenile bees suggests that the virus presents symptoms? And are you saying that it presents symptoms in Vespids or in A. mellifera?
The sentence was changed to “SBV can present causes failure to pupate and subsequent death in A. mellifera, affecting both larvae and adult honey bees [103].”
Lines 486-488 – You mention hornets here because to this point you’ve only talked about their presence in the genus Vespula. Your logic for suggesting that the virus is passed from hornets to honey bees is unclear. ;Lines 489-490 – V. pensylvanica is not a hornet but your language here suggests it is. Please revise for clarity and accuracy.; Lines 492-494 – suggested by whom? Please revise for clarity and Lines 494-496 – citation needed
The entire paragraph has been changed for clarity.
Lines 508-510 – citation needed
Done
[106] Runckel, C.; Flenniken, M.L.; Engel, J.C.; Ruby, J.G.; Ganem, D.; Andino, R.; DeRisi, J.L. Temporal analysis of the honey bee microbiome reveals four novel viruses and seasonal prevalence of known viruses, Nosema, and Crithidia. PloS one 2011, 6(6), e20656.
Line 514 - It is not clear what the word “both” refers to in this sentence since a long list of virus variants precedes it.
It refers to LSV1 and LSV2.
Line 518 – you used the abbreviation for Apis mellifera filamentous virus without defining it first.
Done
Line 526-527 – citation needed
Done
Line 527-528 – it is not clear what you mean in this sentence. How is monitoring careless? Do you mean that it is “infrequent” or maybe “not executed effectively”? Please revise for clarity and use more specific wording. And please provide a citation to support this statement.
The sentence was changed to “Despite being a widely distributed virus, studies on other possible hosts, such as vespids, are scarce.”
Line 537 – the “v” in velutina should not be capitalized.
Done
Line 543 – What are you referring to as the intestine of the Vespid? Wasps don’t have “intestines” proper
It was changed to gut
Lines 550-553 – citation needed
Done
Lines 558-559 – Your logic here is not clear. What is the link between the high diversity of pathogens in V. velutina and the rapid expansion of the species in invaded ecosystems?
The sentence was changed to “The rapid spread of V. velutina in invaded ecosystems, coupled with its ability to carry a wide range of pathogens, is indicative of the potential risk it may pose to the species with which it interacts, such as honey bees.”
Lines 559-560 - citation needed
Done
Line 565 – reference your table here (Table 1).
Done
Line 574 – Are you referring to Southeast Asia when you say “continent”? SEA is not a continent.
It was changed to “region”.
Line 577 – You say V. velutina is native to SEA but then you reference its native range in China which is not in SEA.
The paragraph was changed to “V. velutina is native to Southeast Asia, being common in southern China [14,15]. From these areas it has been able to invade along two routes, through northeast Asia and Europe. In northeast Asia, it spread through South Korea to Japan [15], where it has been introduced as an invasive species (Figure 2b). The main studies on bee viruses in Asia have been carried out with individuals from China.”
Figure 2 – This figure is unclear. You say image (a) shows the invasive area and (b) shows the native area but (b) also has areas of its invasive area labeled. Please clearly identify the native range. And please clarify whether the colors indicate the viruses listed are present in that area or were simply identified by researchers in that region
The figure shows invaded areas in light blue, while the other colors represent the countries where virus studies on V. velutina have been carried out. The explanation to the figure has been modified for better understanding “In addition, figures also show its distribution as an invasive species in light blue (Invasive state)“
Line 589 – “introduced, non-native species” is a redundant phrase. No need to include all of those words.
The sentence was changed to ”This spread also affects the introduction of non-native species, such as Vespids.”
Line 590-591 – The Vespids are not always able to replicate the virus. You will need to qualify this statement
The sentence was change to “The research discussed in this review demonstrates that there is virus transmission from honeybees to vespids, and that virus replication does occur on certain occasions.”
Line 591-593 – please revise for clarity.
This highlights the need to reformulate the specificity of bee viruses for honey bees and to deepen the scarce knowledge on their spread and infection of new hosts.
Line 594-596 – this statement seems contradictory to the statement in like 590-591. You seem to provide unqualified statements of directionality. It’s confusing to then say that no studies demonstrate directionality.
The sentences have been clarified as follows:
In contrast, there are no studies demonstrating reverse directionality, that is, from the vespids to honey bees. However, the literature highlights the spillover of bee viruses into ecosystems invaded by invasive Vespids. This poses a risk for the spread of bee viruses and considers a gap regarding the future of bee virus control in apiaries.

Reviewer 2 Report
The review by authors Rodríguez-Flores et al. is very timely and appropriate. While we are concerned about pathogen transmission between different bee species, we often inadvertently neglect potential pathogen transmission between pests/predators and bee species. The review sufficiently summarizes the concept and I do not have any major or minor edits to suggest.
Author Response
Reviewer_2
The review by authors Rodríguez-Flores et al. is very timely and appropriate. While we are concerned about pathogen transmission between different bee species, we often inadvertently neglect potential pathogen transmission between pests/predators and bee species. The review sufficiently summarizes the concept and I do not have any major or minor edits to suggest.
The authors would like to thank the reviewer for his comments. This paper aims to highlight the lack of research in the study of virus transmission associated with invasive alien species of vespids with increasing globalization.
Reviewer 3 Report
The topic is of high importance and we thank the researchers for it.
This research is a review article, about Emerging risk of cross-species transmission on bee viruses in the presence of invasive vespid species.
The Vespa, Vespula, and Polistes genera, which are members of the Vespidae family, are highlighted in this bibliographic review of the major bee virus studies.
But very little is understood about the symptomatology or its spread. Beekeeping is being significantly impacted by V. velutina, whose geographic distribution outside of its native habitat is expanding.
This review's important information on bee viruses in invading vespids is pertinent to encouraging more study and assisting in halting the spread of bee viruses in invaded environments.
This data must be collected in order to advance studies on the propagation of bee viruses in ecosystems overrun by invasive vespid species and to stop the decrease of bee populations brought on by bee viruses.
Author Response
Reviewer_3
The topic is of high importance and we thank the researchers for it.
This research is a review article, about Emerging risk of cross-species transmission on bee viruses in the presence of invasive vespid species.
The Vespa, Vespula, and Polistes genera, which are members of the Vespidae family, are highlighted in this bibliographic review of the major bee virus studies.
But very little is understood about the symptomatology or its spread. Beekeeping is being significantly impacted by V. velutina, whose geographic distribution outside of its native habitat is expanding.
This review's important information on bee viruses in invading vespids is pertinent to encouraging more study and assisting in halting the spread of bee viruses in invaded environments.
This data must be collected in order to advance studies on the propagation of bee viruses in ecosystems overrun by invasive vespid species and to stop the decrease of bee populations brought on by bee viruses.
The authors would like to thank the reviewer for his comments. The importance of this study lies in the promotion of new research on virus transmission among invasive alien vespid species, which is increasing due to increasing globalization, especially in the case of V. velutina.
Reviewer 4 Report
The spread of Vespidae species as a threat to beekeeping is widely recognized by the beekeepers in several countries and the aspect of viral transmission is potentially highly relevant however yet poorly understood. The review by Rodríguez-Flores summarizes the available literature and discusses the important implications and is an interesting and highly enjoyable read. Therefore, I would recommend to publish it in Insects.
I have some minor suggestions.
1. It should be made clear if the review focuses on viruses of A.mellifera, the economically most important bee species, or if viruses of A.cerana are also meant, viruses of the whole Apis genus, all social bees, or even all bees. If there is such a focus, it should be reflected in the title.
2. The hierarchy in chapter 2. is confusing. "2.1 Types of viral transmission" does not fit well under the chapter name "2. Vespid species as hosts of bee viruses". The easiest solution would be to upgrade 2.1. and 2.2. to chapters 3 and 4. Alternatively, "2. Vespid species as hosts" can be downgraded to 2.1 and an overarching heading must be found, covering all three sections.
3. The way Genus in species names is abbreviated appears sometimes inconsistent (e.g. L. 305+306, L. 345, L. 385), and not ideal, overall. I assume, the authors wanted to the abbreviate the about 10 most common species names and write the others out, which is arbitrary and not every reader agrees with what the more common species is. Therefore, I would recommend to apply a more generic way to abbreviate Genus names, e.g. to abbreviate every species which is mentioned more than once in the text. The complication is that V. stands for Vespa, Vespula and Varroa, however, even the abbreviated species names do not overlap. I would therefore recommend to use the paragraph L 59-73 also as an introductory section for all Wasp species mentioned in the manuscript. In this regard, I would split table 2 and have a separate table for Polistes. In this way, the tables serve as a dictionary for species names.
4. CBPV is missing in Table 1, where citations 19+23 are suitable.
5. Also SBPV is missing in Tables 1 and 2.
Minor issues by line
L 284. I did not understand this sentence, at least not as reasoning for the previous sentence.
L 310. Here I felt that this phrase was repeated one too many.
L 415-416. 100 particles per what?
L 507. When being very accurate about wasp and virus taxonomy, bee species should also get some care. At least they should be "samples of Western honey bee", if not A. mellifera. In USA it might be exclusive, but generally there are 7 other Apis species, and Melipona may be called honey bee as well.
L 586. They "contribute" but they are not solely responsible. There are many other factors.
L 629. preventing the spread of viruses is a bit too much to ask from just understanding the interactions better, and too crude on the other hand. In fact, there a whole lot of other things that can be done with better knowledge: optimizing management, treating endangered colonies, improved diagnostics, targeted measures etc.
Language issues
L. 39 comma should be removed
L. 283 the colony collapses, not the hive
L. 306 solitude "Vespa" - what is meant?
L. 321 No accent on V. germanica (it's Latin)
L. 330 Word order is not correct
L. 592 grammar - a definition or statement can be reformulated, not a virus
L. 594 grammar. I recommend replacing "directionality" with "transmission" and would clarify that the studies did not directly demonstrated it but some deduced it indirectly.
L. 621. "Most" studies? people?
L. 622. symptomatology - the word is a bit over the top when they might have no symptoms at all
L. 620-622. These sentences should be formulated more carefully.
L. 628 "impactful" does not fit here, grammatically
L. 629 better "help to prevent"
Author Response
Reviewer_4
The spread of Vespidae species as a threat to beekeeping is widely recognized by the beekeepers in several countries and the aspect of viral transmission is potentially highly relevant however yet poorly understood. The review by Rodríguez-Flores summarizes the available literature and discusses the important implications and is an interesting and highly enjoyable read. Therefore, I would recommend to publish it in Insects.
I have some minor suggestions.
Thank you very much for your comments and suggestions. Below, each of them are going to be answered to improve the quality and presentation of the manuscript.
- It should be made clear if the review focuses on viruses of A.mellifera, the economically most important bee species, or if viruses of A.cerana are also meant, viruses of the whole Apis genus, all social bees, or even all bees. If there is such a focus, it should be reflected in the title.
The viruses discussed in this study have been determined in both species. Some of them even in Vespid species. Although most of them have been proclaimed as threats to beekeeping, we have modified the title and emphasized that they are honey bee viruses.
“Emerging risk of cross-species transmission of honey bee viruses in the presence of invasive vespid species”
- The hierarchy in chapter 2. is confusing. "2.1 Types of viral transmission" does not fit well under the chapter name "2. Vespid species as hosts of bee viruses". The easiest solution would be to upgrade 2.1. and 2.2. to chapters 3 and 4. Alternatively, "2. Vespid species as hosts" can be downgraded to 2.1 and an overarching heading must be found, covering all three sections.
Done
- The way Genus in species names is abbreviated appears sometimes inconsistent (e.g. L. 305+306, L. 345, L. 385), and not ideal, overall. I assume, the authors wanted to the abbreviate the about 10 most common species names and write the others out, which is arbitrary and not every reader agrees with what the more common species is. Therefore, I would recommend to apply a more generic way to abbreviate Genus names, e.g. to abbreviate every species which is mentioned more than once in the text. The complication is that V. stands for Vespa, Vespula and Varroa, however, even the abbreviated species names do not overlap. I would therefore recommend to use the paragraph L 59-73 also as an introductory section for all Wasp species mentioned in the manuscript. In this regard, I would split table 2 and have a separate table for Polistes. In this way, the tables serve as a dictionary for species names.
Information on the nomenclature of the most described species has been added. In addition, each of the tables has been separated according to the genus in question, which favors the understanding of the species studied.
- CBPV is missing in Table 1, where citations 19+23 are suitable.
We have included the citation [23], since the results of reference [19] mention" No amplicons were detected for SBV, IAPV, ABPV, CBPV, SPV major and SPV minor in all V. velutina specimens.". For this reason, it was not included in Table 1.
- Also SBPV is missing in Tables 1 and 2.
Slow Bee Paralysis Virus (SBPV or SPV) was evaluated by the study of Mazzei et al [19]. However, no amplicon was found.
Minor issues by line
L 284. I did not understand this sentence, at least not as reasoning for the previous sentence.
Authors have clarified the sentence
L 310. Here I felt that this phrase was repeated one too many.
The sentence has been deleted as it has already been mentioned in previous paragraphs.
L 415-416. 100 particles per what?
It refers to100 particles of virus (ABPV). We have changed the expression to clarify this term.
L 507. When being very accurate about wasp and virus taxonomy, bee species should also get some care. At least they should be "samples of Western honey bee", if not A. mellifera. In USA it might be exclusive, but generally there are 7 other Apis species, and Melipona may be called honey bee as well.
The reviewer is right, in addition to the study, written below, makes this distinction, mentioning Western honey bees (Apis mellifera)
Runckel, C., Flenniken, M. L., Engel, J. C., Ruby, J. G., Ganem, D., Andino, R., & DeRisi, J. L. (2011). Temporal analysis of the honey bee microbiome reveals four novel viruses and seasonal prevalence of known viruses, Nosema, and Crithidia. PloS one, 6(6), e20656.
L 586. They "contribute" but they are not solely responsible. There are many other factors.
The sentence was changed to “Bee viruses represent a threat to the loss of honey bee populations.”
L 629. preventing the spread of viruses is a bit too much to ask from just understanding the interactions better, and too crude on the other hand. In fact, there a whole lot of other things that can be done with better knowledge: optimizing management, treating endangered colonies, improved diagnostics, targeted measures etc.
The sentence was changed to “The information impactful in this review on honey bee viruses in invasive Vespids is relevant to promote further research on honey bee virus transmission in vespids and improve a management of infectious diseases caused by bee viruses in apiaries.”
Language issues
L. 39 comma should be removed
Done
L. 283 the colony collapses, not the hive
Done
L. 306 solitude "Vespa" - what is meant?
It was referring to Vespa sp.
Vespa sp.
L. 321 No accent on V. germanica (it's Latin)
Done
L. 330 Word order is not correct
The sentence was changed to “If the queen were to complete her development, she would darken, showing a dark-brown coloration”
L. 592 grammar - a definition or statement can be reformulated, not a virus
L. 594 grammar. I recommend replacing "directionality" with "transmission" and would clarify that the studies did not directly demonstrated it but some deduced it indirectly.
The sentence was changed to “The research discussed in this review demonstrates that there is virus transmission from honeybees to vespids, and that virus replication does occur on certain occasions.”
L. 621. "Most" studies? people?
It refers “Most of the research”
L. 622. symptomatology - the word is a bit over the top when they might have no symptoms at all
Totally agree, many of the viruses discussed in this study do not present symptoms.
L. 620-622. These sentences should be formulated more carefully.
The sentences were changed to “Knowledge about bee viruses in vespid species is scarce. Most of the investigations describe the presence and/or replication of the bee virus in vespids. However, very little is known about the transmission mechanisms of these viruses.”
L. 628 "impactful" does not fit here, grammatically
Done. It was changed to “collected”
L. 629 better "help to prevent"
Done
